# Trend in Breakfast Consumption among Primary School Children in Italy

**DOI:** 10.3390/nu15214632

**Published:** 2023-10-31

**Authors:** Silvia Ciardullo, Michele Antonio Salvatore, Donatella Mandolini, Angela Spinelli, Mauro Bucciarelli, Silvia Andreozzi, Marta Buoncristiano, Paola Nardone

**Affiliations:** 1National Centre for Disease Prevention and Health Promotion, Italian National Institute of Health (Istituto Superiore di Sanità), Viale Regina Elena 299, 00161 Rome, Italy; silvia.ciardullo@iss.it (S.C.); micheleantonio.salvatore@iss.it (M.A.S.); donatella.mandolini@iss.it (D.M.); angela.spinelli@iss.it (A.S.); mauro.bucciarelli@iss.it (M.B.); silvia.andreozzi@iss.it (S.A.); 2“OKkio alla SALUTE” Technical Committee, 00161 Rome, Italy; marta.buoncristiano@gmail.com

**Keywords:** children, breakfast, trend

## Abstract

Eating breakfast daily improves cognitive function, may contribute to learning and academic performance in children and can decrease the risk of childhood obesity. The aim of this study was to analyse how breakfast consumption changed in the period from 2008/9 to 2019 in Italy among children aged 8–9 years old participating in the OKkio alla SALUTE surveys and to explore the associations with some socio-demographic and lifestyle factors. Information about children’s daily breakfast consumption (adequate, inadequate, and no breakfast) and the socio-demographic characteristics of the children and their mothers was collected through four questionnaires addressed to parents, children, teachers and head teachers. Data were analysed for the 272,781 children from 21 Italian regions from 2008/9 to 2019. The prevalence of adequate breakfast decreased from 60.7% in 2008/9 to 55.7% in 2019 and no breakfast from 10.9 to 8.7%; conversely, inadequate breakfast increased from 28.4 to 35.6%. Logistic regression models showed that the occurrence of inadequate or no breakfast consumption was significantly higher among girls, children living in the southern regions and children with less educated mothers. These findings highlight the need for effective interventions to increase daily breakfast consumption and its adequacy among children.

## 1. Introduction

Breakfast is an important meal of the day [1]. Eating breakfast daily improves cognitive function, may contribute to learning and academic performance in children and adolescents [2,3] and can decrease the risk of childhood obesity [4].

Children who consume breakfast regularly are more likely to have favourable macro- and micro-nutrient intakes, including a higher intake of dietary fibre and total carbohydrates, and lower total fats [5].

Conversely, breakfast skipping has been shown to be associated with various unhealthy habits such as engaging in less physical activity and more screen time, having less nocturnal sleep time and poorer dietary habits overall [6,7]. These habits may be maintained from youth into adulthood, so prevention should begin at an early age [8,9].

Several socio-demographic factors can influence breakfast consumption among children and adolescents: age, sex, lower socio-economic status as well as parental education [10]. Furthermore, breakfast consumption rates and meal composition vary significantly between geographical regions and countries [11]. Data from the WHO European Childhood Obesity Surveillance Initiative (COSI) round 5, that involved 29 study locations, showed that on average, almost 75% of children (6–9 years old) had breakfast every day with no important difference between girls and boys. Parental education appeared to have a more consistent impact in nearly all countries, with children of highly educated parents more likely to have breakfast every day [12]. In particular, high maternal educational level is strongly associated with healthy food among children and daily breakfast consumption; infants of mothers who have low levels of education have a higher intake of sugar, fat, and protein [13].

Other evidence suggests that breakfast skipping is most prevalent among females, older children, and adolescents [14,15,16].

The consumption and meal composition of breakfasts among children, the time trend, socio-economic and geographical differences of breakfast habits are important information for planning healthy intervention strategies for target groups.

Since 2007, following the recommendations of the WHO European Ministerial Conference on Counteracting Obesity [17], the Italian Ministry of Health has promoted and funded the National Nutritional Surveillance System called “OKkio alla SALUTE” to obtain information on nutritional status by routine measurement of bodyweight, height and lifestyle behaviour among primary school children [18].

The aim of this study was to analyse how breakfast consumption had changed in the period from 2008/9 to 2019 in Italy among children aged 8–9 years old participating in the OKkio alla SALUTE surveys and to explore the associations with some socio-demographic and lifestyle factors.

## 2. Materials and Methods

### 2.1. Data

Data from the Italian National Surveillance System “OKkio alla SALUTE”, coordinated by the Istituto Superiore di Sanità—Italian National Institute of Health (ISS), were used. OKkio alla SALUTE is a population-based surveillance, based on epidemiological surveys repeated at regular intervals on representative samples of the study population in all 21 Italian regions. The target population is children in the third grade of primary schools, who are almost all 8 or 9 years old. Through OKkio alla SALUTE, Italy participates in the data collection for COSI in the European Region of the WHO, which allows comparable information on children’s nutritional status and lifestyle behaviours to be obtained by the countries participating in the initiative.

The sampling method chosen is a stratified cluster survey design, with classes as the sampling unit, a method recommended by the WHO [19]. Sample selection was carried out in each region at the local health unit (LHU) level from sampling lists of primary third classes provided by the regional school authorities. All children of the selected classes were invited to participate. Regions decide whether to select a sample that is representative of the region or of individual LHUs. The sample size was identified based on the prevalence of the excess weight found in the previous data collection, with a precision of the estimate of 3% for the region and 5% for the LHU. The sample size estimate was also adjusted by assuming a design effect based on the results of the previous round. Further details of the methodology are provided elsewhere [19,20].

This study used data from all rounds of the Surveillance (years 2008/9, 2010, 2012, 2014, 2016, 2019), each involving about 2400 schools and 50,000 children.

According to the COSI protocol [21], children were weighed and measured with standard equipment (Seca 872TM scales and Seca 214 stadiometers. Seca GmbH & Co., Hamburg, Germany) by trained staff of the LHUs conducting the survey. The same procedures were used for all rounds of data collection.

Information on eating habits, physical activity and sedentary behaviours of children was collected through four questionnaires, one for parents and distributed along with the introductory letter and consent form, another for children (distributed with the help of teachers), one for the teachers and the last for the head teachers.

Children were measured with clothes and then the weight was corrected for the estimated weight of the clothes.

On the same day, children were weighed/measured and completed the self-questionnaires. For each round of OKkio alla SALUTE, the data collection was conducted in the same period: from January/February to June. This choice allowed for the comparison of data across the years.

### 2.2. Outcomes

Breakfast consumption (adequate, inadequate, and no breakfast) was the outcome variable in this study. Children were asked if they had had breakfast on the morning of the interview and to indicate the foods they had eaten.

In particular, this section of the questionnaire contained a list of foods that children indicated as foods eaten for breakfast on the morning of the survey (Children’s questionnaire–breakfast section in Appendix A). According to the qualitative approach, the data cleaning and analysis allowed us to understand the type of food consumed, define its quality and describe breakfast habits across the years.

According to the recommendations of the Centre for Research for Food and Nutrition (CREA), breakfast was classified “adequate” when it included proteins and both complex and simple carbohydrates [22].

### 2.3. Covariates

The following socio-demographic and lifestyle factors were included in the study: child’s sex, weight status (classifying children as ≤normal weight, overweight or obese on the basis of their calculated body mass index according to the age and sex-specific WOF-IOTF cut-offs [23,24]), mother’s educational level (low = less than high school, medium = high school, high = university degree or more), mother’s citizenship (Italian, not Italian) and area of residence (north, centre and south of Italy). In the first rounds, the family questionnaire collected data (educational level, citizenship) only for respondents who usually were children’s mothers. For trend analysis, only mothers’ information was considered.

### 2.4. Statistical Analysis

Frequency distributions, prevalence and odds ratios (OR) with 95% confidence intervals (95% CI) were used to describe the data. Where data were missing, percentages were calculated based on cases with known information. STATA/MP version 15 was used to perform statistical analysis.

Frequency distributions of children by outcome and covariates were calculated for each round of data collection. Changes in the occurrence of inadequate or no breakfast consumption were assessed using a multivariate logistic regression model for complex sample survey design where the information on the round of data collection was included as a categorical variable. The socio-demographic and lifestyle factors were included in the model in order to calculate adjusted ORs with 95% CI. Furthermore, the time trends in the prevalence of adequate, inadequate consumption and no breakfast were reported, and the linearity of the trends was tested through separate multivariate logistic regression models including a linear term for time, as well as the other variables included in the study. The trends of the prevalences of breakfast consumption (adequate, inadequate, and no breakfast), relative changes (differences between the last value of the trends and the first value divided by the first, expressed as percentages) and the tests for linear trend through multivariate logistic regression models were also calculated, stratifying by the socio-demographic and lifestyle factors. A significance level of 0.05 was adopted in all models.

## 3. Results

Out of the total 307,953 children in the selected classes in the six rounds of the survey, the data were analysed for the 272,781 children aged 8 or 9 whose parents gave consent for them to participate and were at school on the day of the interview and compiled their questionnaire.

Table 1 shows the main characteristics of the sample by the year of data collection. Overall, 51.4% of children were male; the percentage of overweight children was 21.8% decreasing from 23.2% in 2008/9 to 20.4% in 2019 while the percentage of children with obesity reduced from 12.0% in 2008/9 to 9.4% in 2019 (10.3% overall). Overall, the children of mothers with a low level of education were 32.7%, medium level 47.0% and high level 20.3%. As in the general female population, the educational level of mothers increased from 2008 to 2019 (from 14.2% with high educational level to 28.1%, respectively). The percentage of mothers with foreign citizenship was 12.0%. By design, the distribution by area of residence of the sample was the same as the population of Italy: more than four out of ten children were resident in the north (43.0%), 34.9% in the south and the remainder in the centre (22.1%).

The prevalence of adequate breakfasts decreased from 60.7% in 2008/9 to 55.7% in 2019 (*p* for linear trend < 0.001) and no breakfast from 10.9 to 8.7% (*p* = 0.400) (Figure 1); conversely, the prevalence of inadequate breakfasts increased from 28.4 to 35.6% (*p* < 0.001).

Table 2 shows the adjusted ORs of “bad” breakfast habits (inadequate breakfast consumption or no breakfast) versus adequate breakfast consumption. The time trend of the ORs indicated an increasing occurrence of bad breakfast habits between 2012 (OR = 1.05, 95% CI: 1.00–1.09) and 2019 (OR = 1.35, 95% CI: 1.30–1.41) with respect to the reference round of 2008/9. The test for the linear trend was statistically significant (*p* < 0.001).

The occurrence of inadequate or no breakfast consumption was significantly higher among girls (OR = 1.09, 95% CI: 1.06–1.11), overweight (OR = 1.10, 95% CI: 1.07–1.13) and obese children (OR = 1.26, 95% CI: 1.21–1.30), those with less educated mothers (OR = 1.55, 95% CI: 1.51–1.60) and those living in the southern regions (OR = 1.26, 95% CI = 1.22–1.29).

Table 3 shows the prevalence of breakfast consumption (adequate, inadequate and no breakfast) by socio-demographic and lifestyle factors between 2008/9 and 2019. An increasing trend in inadequate breakfasts was found for all categories of children with relative changes ranging between +11.8% (children with obesity) and +32.0% (children of the most educated mothers). In all cases, the tests for the linear trend were statistically significant. A decrease in no breakfast was observed for all categories of children. The tests for the linear trend were not statistically significant except for children with obesity (relative change −22.0%, *p* for linear trend = 0.015). Adequate breakfast was characterized by a statistically significant decreasing linear trend for all categories (with relative reductions ranging between −12.7% for the north of Italy and medium level of mother’s education and −5.7% for the south of Italy) except for children with obesity (+0.8%, *p* = 0.199). In most cases, this decrease was particularly evident in the correspondence of the last two survey years.

## 4. Discussion

This study investigated the time trend of the consumption and the quality of the breakfasts of children in Italy according to the socio-demographic characteristics of the children and their mothers. The data were large nationally representative samples with a total of 272,871 children.

The results of this study showed that in Italy the proportion of children who consumed an adequate breakfast decreased from 60.7 to 55.7% between 2008/9 and 2019 (about 0.5% per year), whereas the proportion of inadequate breakfasts increased (from 28.4 to 35.6%). In addition, the percentage of children who did not eat breakfast declined from 10.9 to 8.7% in the same period. Compared with other countries participating in the last COSI round, the proportion of children skipping breakfast in Italy was among the lowest [12].

It is important to consider the nutritional quality of breakfast. To be considered good quality, some authors suggest that breakfast should contain three of the main food groups: dairy products, cereals, and fruits [25]. In this study, breakfast was classified as “adequate” when it included proteins and both complex and simple carbohydrates, according to the recommendations of the Italian Centre for Food and Nutrition (CREA) [22]. Our results found that, although about 90% of children consumed breakfast, an important percentage of children did not consume an adequate breakfast and this increased across the years.

The definition of “adequate breakfast” adopted in our study did not allow us to compare our results with other countries; this is a general problem because the definition of breakfast differs between countries and international agreement is necessary to define it, as reported by Gibney et al. [26]. The International Breakfast Research Initiative (IBRI) proposed a harmonised approach to study the nutritional impact of breakfast and to define recommendations for a balanced breakfast, by involving national dietary survey data from six countries [27].

Several studies have found that consuming an inadequate breakfast or no breakfast may have adverse effects on physical and mental performance and on overall health [28,29].

Our data showed differences in the occurrences of inadequate or skipped breakfasts related to the socio-demographic characteristics of the children and their mothers. They were significantly more prevalent among girls, children whose mothers had a low level of education, and among overweight and obese children and those living in the Southern Italian regions. These findings are consistent with other studies carried out worldwide among schoolchildren [12,30,31].

Eating patterns and food preferences in childhood are shaped by individual, interpersonal, and environmental factors, including the child’s family structure, cultural background, social environment, socioeconomic status, and school environment [32,33]. In particular, parents and caregivers can establish a healthy model for their children by helping them to have a healthy diet [34,35]. Several studies have shown that a high educational level of the mother is positively associated with healthy food choices and a healthy lifestyle among children. Probably, high maternal educational level implies better nutritional knowledge, food choices, and parenting practices [34]. Our results found a strong association between mothers’ educational level and the consumption of an adequate breakfast among children. There was an increased prevalence of adequate breakfast consumption found in children from low to high levels of education of the mother. It also showed the same association for children who claimed to skip breakfast; decreasing levels in skipping this meal were observed from low to high educational levels of the mother.

Our results are also consistent with other Italian papers that showed a high prevalence of overweight and obese children in the southern regions, as well as a large diffusion of unhealthy habits [18].

In Italy, despite numerous recent interventions promoting healthy lifestyles at regional and local levels—such as improving eating habits and increasing physical activity in schools—unhealthy behaviours among children persist [36,37].

In particular, the increase in the numbers of children having inadequate breakfasts from 2008/9 to 2019 underlines that more action should be taken to promote a healthy lifestyle in young people and their families. Furthermore, our results indicated that the socio-demographic gap in children’s eating habits across the years persists; this finding marks the importance of focusing the interventions on the most disadvantaged families.

The challenge is to introduce policies at school and community level to involve children and their parents in focused programs about nutritional education. Healthcare professionals should support the parents and children through simple and clear information about healthy lifestyle at an early age.

Further research is needed to understand if and how the COVID-19 pandemic might have changed the eating habits of children and their families as well as increasing the socio-demographic gap between children. It would be interesting to monitor these possible changes by comparing pre- and post-COVID-19 data from the OKkio alla SALUTE surveillance system.

### Limitations and Strengths of the Study

The main strength of this study was the use of large and nationally representative samples to investigate the time trend of the breakfast habits and the associations with socio-demographic characteristics with a low percentage of missing values. The use of standardised data collection procedures based on the international COSI which allowed us to compare Italian data with those from other involved countries is another strength of this study.

However, this study also had some limitations. First, the questionnaire used in OKkio alla SALUTE does not include information on the portion sizes. Second, we used the definition of “adequate” breakfast recommended by the Italian Centre for Food and Nutrition (CREA), which might not be comparable with the definitions used in other studies. These limitations depended on the methodology of this study: the population surveillance. The OKkio alla SALUTE surveillance system describes the prevalence of the children’s nutritional status and their associated factors (i.e., eating habits, physical activity, etc.) to support national programs promoting healthy child development.

## 5. Conclusions

Our study supplements the current literature on factors that influence breakfast consumption among children. Public health interventions aimed at promoting healthy childhood growth and development should consider the parents’ knowledge and education as well as the sociocultural contexts in which children and families live.

Future interventions should exploit the policies, strategies and programmes adopted in other countries that have experienced an increase in adequate daily breakfast consumption over time. Furthermore, to understand the determinants that influence the decline in adequate breakfast consumption, focused qualitative studies should be undertaken.

## Figures and Tables

**Figure 1 nutrients-15-04632-f001:**
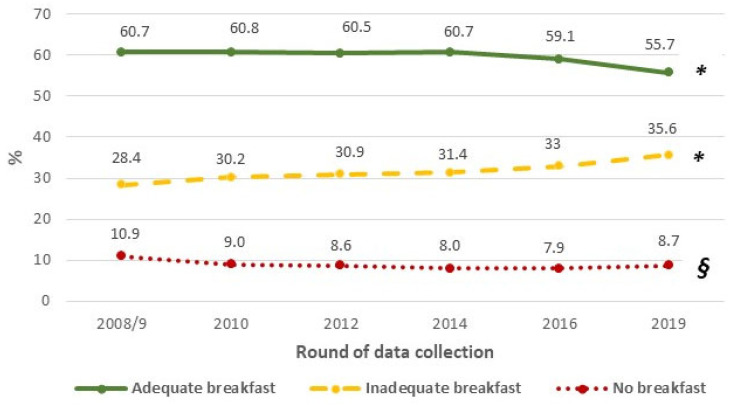
Time trend in the prevalence of breakfast (adequate, inadequate, no breakfast) among primary school children from 2008/9 to 2019. *******
*p* for linear trend < 0.001. ***§***
*p* for linear trend = 0.400.

**Table 1 nutrients-15-04632-t001:** Sample characteristics by round of data collection. Italy, 2008/9–2019.

Variables	Round of Data Collection						
	2008/9	2010	2012	2014	2016	2019	Total
	N = 47,293	N = 41,762	N = 45,518	N = 47,445	N = 44,680	N = 46,173	N = 272,871
	%	%	%	%	%	%	%
**Child’s age**							
8 years	64.1	64.3	66	67.9	66.7	64.8	65.7
9 years	35.9	35.7	34	32.1	33.3	35.2	34.3
**Child’s sex**							
Male	51.6	51.7	50.8	51.7	51.7	51.1	51.4
Female	48.4	48.3	49.2	48.3	48.3	48.9	48.6
*missing*	*0.02*	*0.01*	*0.00*	*0.00*	*0.81*	*0.71*	*0.27*
**Weight status**							
≤normal weight	64.9	65.8	67.2	69.2	69.4	70.2	67.9
overweight	23.2	23.0	22.2	20.9	21.3	20.4	21.8
obese	12.0	11.2	10.6	9.8	9.3	9.4	10.3
*missing*	*0.08*	*0.49*	*0.21*	*0.28*	*1.15*	*1.04*	*0.55*
**Mother’s educational level**							
Low	39.8	38.0	35.1	31.3	28.2	25.1	32.7
Medium	46.0	46.2	46.9	47.9	48.1	46.7	47.0
High	14.2	15.8	18.0	20.8	23.7	28.1	20.3
*missing*	4.0	5.4	*4.9*	4.8	4.9	5.9	5.0
**Mother’s Citizenship *****							
Italian	-	90.7	88.5	87.1	86.2	87.9	88.0
Foreign	-	9.3	11.5	12.9	13.8	12.1	12.0
*missing*	-	*5.0*	*4.5*	*4.5*	*3.7*	*7.2*	5.0
**Area of residence**							
North	34.7	36.3	44.7	47.4	45.5	47.3	43.0
Centre	23.5	23.7	21.1	21.2	21.9	21.9	22.1
South	41.8	40.0	34.2	31.4	32.6	30.8	34.9

******* Information on mother’s citizenship was not gathered in 2008/9.

**Table 2 nutrients-15-04632-t002:** Mutually adjusted odds ratios of inadequate or no breakfast consumption for the reported variables—logistic regression models. Italy, 2008/9–2019.

Variables	OR	95% CI		*p*-Value
**Round of data collection**				
2008/9	1			
2010	1.01	0.97	1.05	0.706
2012	1.05	1.00	1.09	0.035
2014	1.07	1.03	1.12	0.002
2016	1.16	1.11	1.21	<0.001
2019	1.35	1.30	1.41	<0.001
**Linear trend over the period 2008/9–2019**				
	1.03	1.02	1.03	<0.001
**Child’s sex**				
Male	1			
Female	1.09	1.06	1.11	<0.001
**Weight status**				
≤ normal weight	1			
overweight	1.10	1.07	1.13	<0.001
obese	1.26	1.21	1.30	<0.001
**Mother’s educational level**				
High	1			
Medium	1.31	1.28	1.35	<0.001
Low	1.55	1.51	1.60	<0.001
Area of residence				
North	1			
Centre	0.93	0.90	0.96	<0.001
South	1.26	1.22	1.29	<0.001

**Table 3 nutrients-15-04632-t003:** Prevalence of breakfast consumption (adequate, inadequate, no breakfast) among primary school children from 2008/9 to 2019 by gender, weight status and maternal characteristics.

Variables	Round of Data Collection						Relative Change 2019 vs. 2008/9 %	*p*-value for Linear Trend
	2008/9 %	2010 %	2012 %	2014 %	2016 %	2019 %		
**Child’s sex**								
Male								
No breakfast	10.7	9.3	8.7	8.3	7.7	9.0	−15.7	0.909
Inadequate breakfast	27.3	28.9	30.1	30.4	32.5	34.0	24.5	<0.001
Adequate breakfast	62.0	61.8	61.2	61.3	59.8	57.0	−8.1	<0.001
Female								
No breakfast	11.0	8.7	8.5	7.7	8.1	8.3	−24.4	0.158
Inadequate breakfast	29.6	31.7	31.8	32.4	33.6	37.1	25.3	<0.001
Adequate breakfast	59.3	59.7	59.7	60.0	58.3	54.5	−8.1	<0.001
**Weight status**								
≤Normal weight								
No breakfast	8.7	7.2	7.2	6.6	6.7	7.3	−16.1	0.361
Inadequate breakfast	28.3	29.9	30.5	31.2	33.2	36.0	27.2	<0.001
Adequate breakfast	63.0	62.9	62.3	62.2	60.1	56.7	−10.0	<0.001
Overweight								
No breakfast	13.0	11.4	10.0	9.6	9.3	10.7	−17.7	0.197
Inadequate breakfast	27.6	30.6	31.5	31.3	32.5	34.7	25.7	<0.001
Adequate breakfast	59.4	58.0	58.6	59.1	58.2	54.6	−8.1	<0.001
Obese								
No breakfast	18.6	14.4	14.5	14.2	13.2	14.5	−22.0	0.015
Inadequate breakfast	30.5	31.0	32.5	32.3	32.9	34.1	11.8	<0.001
Adequate breakfast	51.0	54.6	53.0	53.4	53.8	51.4	0.8	0.199
**Mother’s educational level**								
Low								
No breakfast	13.4	11.2	11.5	10.6	11.2	11.4	−14.9	0.121
Inadequate breakfast	29.9	32.7	33.2	33.6	34.5	36.3	21.4	<0.001
Adequate breakfast	56.6	56.1	55.2	55.8	54.2	52.3	−7.6	<0.001
Medium								
No breakfast	9.7	8.2	7.6	7.4	7.5	8.9	−9.0	0.880
Inadequate breakfast	28.2	29.4	31.0	31.7	33.4	37.0	31.2	<0.001
Adequate breakfast	62.0	62.4	61.4	60.9	59.0	54.1	−12.7	<0.001
High								
No breakfast	6.4	4.8	4.5	4.8	4.7	5.4	−15.7	0.920
Inadequate breakfast	25.0	26.7	26.2	27.5	30.5	33.0	32.0	<0.001
Adequate breakfast	68.6	68.4	69.3	67.7	64.9	61.7	−10.1	<0.001
**Area of residence**								
North								
No breakfast	7.0	5.5	5.8	5.1	5.2	6.3	−10.8	0.811
Inadequate breakfast	29.0	30.4	31.3	32.1	34.1	37.8	30.3	<0.001
Adequate breakfast	64.0	64.1	62.9	62.8	60.8	55.9	−12.7	<0.001
Centre								
No breakfast	9.4	8.5	8.1	6.9	8.0	7.6	−18.8	0.269
Inadequate breakfast	25.9	28.1	28.0	27.2	30.4	32.4	25.1	<0.001
Adequate breakfast	64.7	63.4	63.9	65.9	61.6	60.0	−7.3	<0.001
South								
No breakfast	14.9	12.5	12.6	13.0	11.7	13.2	−11.4	0.495
Inadequate breakfast	29.4	31.3	32.2	33.0	33.2	34.4	17.0	<0.001
Adequate breakfast	55.7	56.2	55.2	54.0	55.1	52.5	−5.7	<0.001

## Data Availability

OKkio alla SALUTE data and questionnaires can be accessed via a request to the Principal Investigator, Dr. Paola Nardone: paola.nardone@iss.it. For further information, https://www.epicentro.iss.it/okkioallasalute/(accessed on 20 September 2023).

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
