# Peer review of "Trend in Breakfast Consumption among Primary School Children in Italy"

_nutrients, 2023, doi:10.3390/nu15214632_

Round 1
Reviewer 1 Report
Comments and Suggestions for Authors
The study highlights the importance of daily breakfast consumption in children and its positive effects on cognitive function, learning, academic performance, and obesity prevention. The study analyzed the changes in breakfast consumption among children aged 8-9 years old in Italy from 2008/9 to 2019 and explored the associations with socio-demographic and lifestyle factors. The study found that the prevalence of adequate breakfast decreased, while inadequate breakfast increased, and that girls, children living in Southern regions, and children with less educated mothers were more likely to have inadequate or no breakfast consumption. The study suggests the need for effective interventions to increase daily breakfast consumption and its adequacy among children.
One of the main conditions for analyzing breakfast is precisely the concept of adequacy. Based solely on the consumption of protein and carbohydrates, breakfast can be highly incorrect from a nutritional point of view, such as a combination of sausages and a sweetened drink. This would meet the criterion used in the study as adequate, but it certainly can't be considered so from a nutritional point of view.
Numerous classifications or nomenclatures exist in the literature on the adequacy of breakfast, with definitions based on energy volume, the combination of foods (dairy, carbohydrates, and fruit), the minimum value of 50 calories, or the frequency of consumption.
One of the examples that should be analyzed is the IBRI, or International Breakfast Research Initiative, initially led by Prof. Gibney. Despite possible industrial interests, worldwide scientific production is extensive and robust. It evaluates not only the existence of breakfast but also its composition, volumes ingested, time and schedule of consumption, and comparative nutritional indices. Some of the internationally published papers are available in this compilation: https://www.mdpi.com/journal/nutrients/special_issues/breakfast_research
The Italian study makes virtually no reference to other studies on populations and only refers to the country's concept. They recognize the limitation of the work that it has problems for international comparison due to the use of a single criterion. There is no consumption data, no description of the food, the type of protein or carbohydrate, the frequency of consumption, or the volume of food ingested.
The use of questionnaires in children aged 8 and 9 is complex, and the questionnaire needs to be presented. There is a possible confusion in the work, which is the lack of description of WHEN the questionnaires and anthropometric analysis were carried out. As there is only a numerical description, it is assumed that they were carried out during the last assessment in 2019. There is doubt about this final data since the beginning of 2019 marks the start of the COVID pandemic and the closure of all schools, especially in Italy. When was the questionnaire administered since it was apparently in person?
The mathematical analysis is correct, but there is no indication that the same questionnaires and anthropometric assessment were used in all the years of application. The methodology needs to be more adequately described.
The comparison of data with other populations ends up needing to be revised since different and non-comparable criteria are used. There are several studies on the subject in the European community itself that still need to be cited.
Author Response
Response to Reviewer 1 Comments
Dear reviewer,
many thanks for your review of our manuscript. We revised the manuscript by modifying the sections based on your suggestions and comments. Our case-by-case responses to the reviewer suggestion and comments were also provided. Revisions in the text are in red. We hope that the revised manuscript will now be deemed acceptable for publication in the International Nutrients-Special Issue "Dietary and Nutritional Status Assessment in Children and Adolescents in European Countries".
Comments and Suggestions for Authors
The study highlights the importance of daily breakfast consumption in children and its positive effects on cognitive function, learning, academic performance, and obesity prevention. The study analyzed the changes in breakfast consumption among children aged 8-9 years old in Italy from 2008/9 to 2019 and explored the associations with socio-demographic and lifestyle factors. The study found that the prevalence of adequate breakfast decreased, while inadequate breakfast increased, and that girls, children living in Southern regions, and children with less educated mothers were more likely to have inadequate or no breakfast consumption. The study suggests the need for effective interventions to increase daily breakfast consumption and its adequacy among children.
One of the main conditions for analyzing breakfast is precisely the concept of adequacy. Based solely on the consumption of protein and carbohydrates, breakfast can be highly incorrect from a nutritional point of view, such as a combination of sausages and a sweetened drink. This would meet the criterion used in the study as adequate, but it certainly can't be considered so from a nutritional point of view.
Numerous classifications or nomenclatures exist in the literature on the adequacy of breakfast, with definitions based on energy volume, the combination of foods (dairy, carbohydrates, and fruit), the minimum value of 50 calories, or the frequency of consumption.
One of the examples that should be analyzed is the IBRI, or International Breakfast Research Initiative, initially led by Prof. Gibney. Despite possible industrial interests, worldwide scientific production is extensive and robust. It evaluates not only the existence of breakfast but also its composition, volumes ingested, time and schedule of consumption, and comparative nutritional indices. Some of the internationally published papers are available in this compilation: https://www.mdpi.com/journal/nutrients/special_issues/breakfast_research
The Italian study makes virtually no reference to other studies on populations and only refers to the country's concept.
They recognize the limitation of the work that it has problems for international comparison due to the use of a single criterion. There is no consumption data, no description of the food, the type of protein or carbohydrate, the frequency of consumption, or the volume of food ingested.
The use of questionnaires in children aged 8 and 9 is complex, and the questionnaire needs to be presented.
Thank you for your comments. We added some details in the text (line 119-124; 129-133) and we decided to add the children’s questionnaire- breakfast’s section in Appendix 1. Also, please see line 251-257 and 313-316.
There is a possible confusion in the work, which is the lack of description of WHEN the questionnaires and anthropometric analysis were carried out.
As there is only a numerical description, it is assumed that they were carried out during the last assessment in 2019.
Thank you for your suggestion. We added more details in the text: from line 121 to 124.
There is doubt about this final data since the beginning of 2019 marks the start of the COVID pandemic and the closure of all schools, especially in Italy. When was the questionnaire administered since it was apparently in person?
Thank you for your comments. The survey 2019 of OKkio alla SALUTE started on February 2019 and ended on June 2019 (1 year before the COVID-19 pandemic).
The mathematical analysis is correct, but there is no indication that the same questionnaires and anthropometric assessment were used in all the years of application. The methodology needs to be more adequately described.
Thank you, we added the specific requested. Please, see line 113-114.
The comparison of data with other populations ends up needing to be revised since different and non-comparable criteria are used. There are several studies on the subject in the European community itself that still need to be cited.
Thank you for your comments. As you can see, we compared only the percentage of children skipping breakfast with other countries because there is a problem with the comparability of adequate breakfast. Anyway, following your suggestions, we explained this concept in the text (please see line 251-257) in which we mentioned the IBRI initiative that highlighted the necessity to define breakfast at an international level to compare data across countries.

Reviewer 2 Report
Comments and Suggestions for Authors
This manuscript analyses the trend of breakfast consumption in children. The manuscript flows accordingly, is adequately illustrated, and contains updated references. It is suitable for publication in the present form, but it has to be improved with the following minor revisions.
Materials and methods section
-statistical analysis: at what value of P, the level of significance was set? Please, specify.
Results
-Results have to be described according to the order reported in the tables.
-Are there underweight children? They are not considered.
-Why do you consider only the mother’s educational level and not the father’s one or of someone in their stead? And the same consideration for citizenship. In my opinion, even if it has been demonstrated that mainly maternal educational level is strongly associated with daily breakfast consumption, it could be important to consider parental education.
-Adequate or inadequate breakfast was established according to what children ate on the morning of the interview. In my experience, children or adolescents sometimes skipped breakfast, due to lack of time or appetite and a desire to sleep longer in the morning. This is also confirmed by literature studies. A recent systematic review reporting on the prevalence of 477 breakfast skipping among children and adolescents from 33 countries concluded that 478 most studies reported between 10–30% of young people skipped breakfast [Monzani, A.; Ricotti, R.; Caputo, M.; Solito, A.; Archero, F.; Bellone, S.; Prodam, F. A systematic review of the association of skipping breakfast with weight and cardiometabolic risk factors in children and adolescents. What should we better investigate in the future? Nutrients 2019, 11, 387]. You could have classified a child as “having no breakfast” because he/she hadn't had it on the morning of the interview, but he/she could have skipped breakfast only that morning and on the other days he/she has adequate breakfast. How do you consider this aspect (i.e. occasional skipping breakfast?).
Discussion
What could be the reasons of the decrease in an adequate breakfast consumption?
Author Response
Response to Reviewer 2 Comments
Dear reviewer,
many thanks for your review of our manuscript. We revised the manuscript by modifying the sections based on your suggestions and comments. Our case-by-case responses to the reviewer suggestion and comments were also provided. Revisions in the text are in red. We hope that the revised manuscript will now be deemed acceptable for publication in the International Nutrients-Special Issue "Dietary and Nutritional Status Assessment in Children and Adolescents in European Countries".
This manuscript analyses the trend of breakfast consumption in children. The manuscript flows accordingly, is adequately illustrated, and contains updated references. It is suitable for publication in the present form, but it has to be improved with the following minor revisions.
Materials and methods section
-statistical analysis: at what value of P, the level of significance was set? Please, specify.
Thank you for your question. We added the information on the level of significance adopted in the models, please see line 166.
Results
-Results have to be described according to the order reported in the tables.
Thank you for your comment. We changed the description of the results in accordance with the order of the variables reported in the tables, please see lines 174-189 and lines 211-216 (in red).
We changed the order of results in the table 2 in accordance with the description in the text (see table 2 in red).
-Are there underweight children? They are not considered.
Thank you for your question. Yes, there are underweight children in our data (1.3% mean frequency for all rounds of data collection considered in the paper). Underweight data did not included in this paper because we decided to consider the association between skipping breakfast and children overweight, as reported in several studies; moreover, this choice allowed to compare our results with other countries (for example the COSI network).
-Why do you consider only the mother’s educational level and not the father’s one or of someone in their stead? And the same consideration for citizenship. In my opinion, even if it has been demonstrated that mainly maternal educational level is strongly associated with daily breakfast consumption, it could be important to consider parental education.
Thank you for your question and comments. We added more details about this choice in the text, please see line 144-147.
-Adequate or inadequate breakfast was established according to what children ate on the morning of the interview. In my experience, children or adolescents sometimes skipped breakfast, due to lack of time or appetite and a desire to sleep longer in the morning. This is also confirmed by literature studies. A recent systematic review reporting on the prevalence of 477 breakfast skipping among children and adolescents from 33 countries concluded that 478 most studies reported between 10–30% of young people skipped breakfast [Monzani, A.; Ricotti, R.; Caputo, M.; Solito, A.; Archero, F.; Bellone, S.; Prodam, F. A systematic review of the association of skipping breakfast with weight and cardiometabolic risk factors in children and adolescents. What should we better investigate in the future? Nutrients 2019, 11, 387]. You could have classified a child as “having no breakfast” because he/she hadn't had it on the morning of the interview, but he/she could have skipped breakfast only that morning and on the other days he/she has adequate breakfast. How do you consider this aspect (i.e. occasional skipping breakfast?).
Thank you for your comments. Data about breakfast derived from two different questionnaires: child and parents’ questionnaires. The child questionnaire includes questions about habits referred to the morning in which the questionnaire was filled out (to avoid the children recall bias). In contrast, the parents’ questionnaire includes questions about children habits during the week. The consistency check between the two questionnaire responses allowed to assume that children breakfast habits, declared in the morning of the survey, can be extended to weekly habits.
Discussion
What could be the reasons of the decrease in an adequate breakfast consumption?
Thank you for your fruitful suggestion/question. The cross-sectional design of the Okkio alla SALUTE surveillance system precludes any causality statement and we need other studies design (for example qualitative approach) to understand the determinants which influence this children habits. We added in the conclusion section the necessity to undertake this approach (line 324-326).
